# Morale and Perceived Threats as Predictors of Psychological Coping with Distress in Pandemic and Armed Conflict Times

**DOI:** 10.3390/ijerph18168759

**Published:** 2021-08-19

**Authors:** Yohanan Eshel, Shaul Kimhi, Hadas Marciano, Bruria Adini

**Affiliations:** 1Stress and Resilience Research Center, Tel Hai College, Tel Hai 1220800, Israel; yochi_eshel@hotmail.com (Y.E.); shaulkim@telhai.ac.il (S.K.); hmarcia1@univ.haifa.ac.il (H.M.); 2The Institute of Information Processing and Decision Making (IIPDM), The Ergonomics and Human Factors Unit, University of Haifa, Haifa 3498838, Israel; 3Department of Emergency and Disaster Management, School of Public Health, Sackler Faculty of Medicine, Tel Aviv University, Tel Aviv 6139601, Israel

**Keywords:** COVID-19, armed conflict, morale, distress, resilience, well-being, perceived risks, positive and negative cognitive appraisals

## Abstract

The present study investigated predictors of psychological coping with adversity responses during the COVID-19 pandemic and an armed conflict. Two paired samples that represented the Israeli population that was exposed to both adversities were compared. Respondents rated five different psychological coping responses associated with the two adversities, such as anxiety or individual resilience. Perceived security, pandemic, economic, and political risks, as well as level of morale, were rated. Two major findings were disclosed by two path analyses. Morale improved the predictions of the varied coping responses in both the pandemic and conflict and was the best predictor of four out of five responses and the second-best predictor of the fifth response. Contrary to previous studies, our findings revealed that the concept of a single major predictor of coping responses under distress is an overgeneralization. In both cases, the coping responses were better explained by other perceived risks rather than by the risk of the investigated adversity. Rather than assume that a perceived security threat accounts for low levels of public moods, it is vital to study the antecedents of coping responses and to empirically examine additional potential predictors.

## 1. Introduction

Studies conducted primarily in the aftermath of catastrophes tend to emphasize the negative factors that caused negative psychological outcomes. Thus, it was found that radiation exposures, the World Trade Center attacks, oil spills, mass shootings, hurricanes, and floods have been associated with increases in depression, post-traumatic stress disorder (PTSD), substance use, generalized anxiety disorder, and a range of other negative mental health outcomes [1,2,3,4,5]. Large-scale Chinese research on the impacts of the COVID-19 pandemic has reported that the onset of the coronavirus crisis led to a 74% decrease in overall emotional well-being [6]. An Italian study has found that up to 30% of adults and children were at high risk for post-traumatic stress disturbances during the COVID-19 pandemic period [7], and several studies reported an increase in mental illnesses since the coronavirus outbreak [8,9]. The study of Levkovich and Shinan-Altmans [10] reported further that the negative emotional reactions expressed by the general public during the COVID-19 pandemic reflected the perceived threat of this plague. Similarly, a comprehensive review of the psychological effects of armed conflicts on the general public’s mental health concluded that such armed conflicts confronted in many countries, including Afghanistan, Albania, Cambodia, Israel, and Kurdistan, resulted in higher levels of anxiety, depression, and PTSD [11]. Likewise, a study of the psychological costs of the Vietnam War claimed that they extended beyond PTSD and included an increased risk for depression, personality disorders, suicide, and alcohol abuse [12]. 

It is quite evident that catastrophes and adversities of all kinds are likely to increase a variety of distress reactions. However, most of the pandemics and armed conflicts studies concluded or implied that these responses reflected the direct perceived threats of each of these investigated disasters. Most of them did not examine the extent to which these responses were predicted by the perceived threats of the investigated adversity, as compared to other potential predictors. Studies of anxiety in different contexts indicated, for instance, that it was significantly predicted by a large number of psychological and environmental factors such as serious problems at work, domestic violence, unhappy relationships with family, and higher levels of nonorganizational religious activity and intrinsic religiosity [13], as well as by both financial distress and other psychiatric or mental disorders [14]. It was also found that mood fluctuations and anxieties during the COVID-19 pandemic were predicted concurrently by the pandemic threats, as well as by prior mental health status, and lifestyle changes [15]. Gomes et al. [16] found similarly that mental health problems were significantly related to perceptions of threat, sense of control, and previous coping with distressful encounters. An Israeli study conducted on students [17] has found further that negative, as well as positive, psychological responses during the COVID-19 pandemic were predicted concurrently by perceived health, economic, and security threats. 

### 1.1. Cognitive Appraisals and Coping

The concept of cognitive appraisals was derived from Lazarus’s classic appraisal theory of emotion, which defined appraisals as the personal significance of an encounter for well-being and a proximal determinant of emotion generation [18]. Blascovich and Mendes [19] argued that cognitive appraisals can impact affective, physiological, and behavioral responses in distressful situations that require instrumental responding. A large number of cognitive appraisal studies assessed the impacts of threats on coping. Salkovskis et al. [20] have found that individuals’ appraisal of the danger of germ spread was significantly related to emotional and behavioral responses to the transmission of SARS. These responses included anxiety, avoidance, and disgust. A different perspective of cognitive appraisals described the process of evaluating a stimulus as either a challenge to be met or an overwhelming obstacle from which to retreat [21]. Several researchers examined, therefore, the impact of positive cognitive appraisals on coping with different kinds of distress. Litwic-Kaminska [22] has found that when athletes regard a situation as a challenge, they are more likely to feel confident about their ability to control the situation and tend to develop a strong motivation to prepare well for the competition. Likewise, a study of earthquakes and tsunamis concluded that proactive appraisals of victims lowered their levels of depression and distress [23]. These results suggested that positive cognitive appraisals may have a major role in coping with distress. 

Morale is a well-known positive cognitive appraisal that was generally studied in the domain of work. Research has found that morale was the psychological factor that resulted in positive behavior of employees, and this positive behavior results in increased work efforts and effective performance [24,25]. The role of morale as a distress-reducing factor was investigated to some extent in the military context. An analysis of several modern wars [26] concluded that when a military force fostered high morale among its troops, it was less likely to suffer a substantial number of distress casualties. An example of this was the Falklands War, where the morale of the British troops was high and the distress casualty rate was approximately only 4%. However, the Malta campaign of 1942 was associated with low morale among the British troops and resulted in a level of distress casualties that was estimated as at least 25% of the deployed force. Therefore, we suggest that morale can be viewed as a future-oriented perspective regarding the challenges of coping with one’s current situation. A higher level of morale is likely to be associated with a more positive future orientation and with better coping with hard times. It should be noted that the potential role of morale as a distress-reducing element was generally ignored by the research on disasters and catastrophes, as well as the research on the coping responses that were consequently elevated by these events. 

The people of Israel have recently experienced two major calamities: the COVID-19 pandemic and the May 2021 armed conflict between Israel and Hamas in the Gaza Strip. Israeli cities and settlements suffered massive missiles attacks fired from the Gaza Strip, as well as a few similar attacks from Lebanon, which were followed by riots between Arabs and Jews in several parts of the country. Simultaneously, the Israeli Defense Force (IDF) attacked specific targets in both the Gaza Strip and Lebanon. The present study examined the extent to which negative and positive psychological coping responses that were identified during the COVID-19 pandemic and during the May 2021 hostility were predicted by the health, security, economic, and political perceived risks, as well as by the level of individual morale.

### 1.2. Psychological Coping Responses

**Distress**. The outbreak of COVID-19 was negatively associated with psychological distress responses of grief, hopelessness, posttraumatic symptoms, panic attacks, distress, anxiety, depression, loneliness, ambivalence, fear, stigma, and concern towards socioeconomic status [27,28]. A recent review of the research, which summarized 17 articles from different countries [29], further found a high prevalence of distress among the investigated general populations along with the COVID-19 pandemic. These responses were negatively correlated with a sense of well-being and individual, community, and national resilience [30,31,32]. 

**Well-being**. This is the subjective feeling of health and a positive perception of an individual’s quality of life [33]. Well-being was described as a state of complete physical, mental, and social welfare and not merely the absence of disease or infirmity [34]. High positive correlations were found between well-being, happiness, psychological quality of life, life satisfaction, and positive effects [35]. In addition, well-being was positively associated with individual resilience [35].

**Individual resilience**. Individual resilience constitutes a stable trajectory of healthy functioning after a highly adverse event [36]. The American Psychological Association [37] defines resilience as “the process of adapting well in the face of adversity, trauma, tragedy, threats or other significant sources of distress (paragraph 4)”. Under the coronavirus pandemic threat, individual resilience was negatively and significantly correlated with a sense of danger and distress symptoms [38].

**Community resilience**. Community resilience refers to a community’s ability to cope with stressful conditions, such as natural adversities or man-made calamities, and to recuperate after them. Eachus [39] defined community resilience as “the community’s capability to anticipate risk, limit impact, and bounce back rapidly through survival, adaptability, evolution, and growth in the face of turbulent change”. Examination of community resilience of 12 neighborhoods in New York and New Jersey severely affected by Superstorm Sandy indicated that people living in communities with higher social cohesion, informal social control, and social exchange were more likely to believe that their neighborhoods are very well prepared for a disaster [40].

**National resilience**. This concept reflects a successful national adjustment and functioning efficiently following potentially traumatic events [32]. Canetti et al. [41] have claimed accordingly that national resilience should be defined as the nation’s ability to cope successfully with its disasters (such as poverty, terrorism, or corruption) while keeping its social fabric intact. National resilience correlated negatively with distress symptoms and a sense of danger and correlated positively with a sense of coherence [31].

**Morale**. The concept of morale, which originated in a military context [42], is a multifaceted, longitudinal, and relational experience that individuals share when they identify with and contribute to certain kinds of collective activities [43]. Morale is regarded by Weakliem and Frenkel [25] as a general term for positive feelings about the prescribed activities of the group. The level of morale was found to correlate negatively with a sense of danger and depression and positively with individual and national resiliencies [32].

Considering the importance of identifying predictors of psychological coping with adversities, the current study aimed to compare coping responses during two different coexisting adversities in Israel: the COVID-19 pandemic versus the May 2021 armed conflict between Israel and the Hamas in the Gaza Strip.

### 1.3. Hypotheses

The following hypotheses were investigated: (1) Morale, which constitutes a positive cognitive appraisal, will be as good a predictor as the four investigated negative cognitive appraisals (health, economic, political, and security risks) in predicting both positive and negative psychological coping responses, in both stressful times of the COVID-19 pandemic and 21 May hostility. Higher morale will predict lower levels of distress, and higher levels of each one of the four positive coping responses (well-being and individual, community, and national resilience). (2) In line with a previous study that found that different perceived risks contributed to predicting psychological coping responses during the COVID-19 pandemic (Eshel et al., submitted), we hypothesized that the perceived health risk of the pandemic will not be the best predictor of each of the five psychological coping responses expressed during the COVID-19 pandemic, and some of them will be better predicted by the other perceived risks, less directed to the health lineament of the pandemic. By the same token, the perceived security risk will not be the best predictor of each of these responses expressed immediately after the May 2021 armed conflict. 

## 2. Materials and Methods

### 2.1. Settings

The study measured the variables during two periods of time—the COVID-19 pandemic and a security conflict. The COVID-19 pandemic started in Israel upon the initial identification of confirmed cases in February 2020. It continued in three main waves and substantially receded at the beginning of 2021, following a successful vaccination campaign. By 19 April 2021, 88% of individuals from the age of 50 years or higher were vaccinated after receiving two doses [44]. During June 2021, a fourth wave of COVID-19 began, and subsequently, an additional campaign was launched to inoculate specific risk groups (50 years old and above as well as individuals who are immunocompromised) with a third vaccine (“booster”). 

The violent hostility clash between Israel and Hamas in the Gaza Strip erupted on 10 May 2021 and lasted for 11 days, characterized by massive rocket attacks against civilian communities in the central and the southern parts of Israel. In addition, few rocket attacks were also aimed at Northern Israel, from Lebanon. During that period, domestic violent riots between Arabs and Jews also spread in many areas of Israel (the rural Galilee and many mixed cities in Israel, such as Lod, Acre, and Jaffa). 

### 2.2. Participants

The data for the current study were collected via an internet panel company possessing a database of more than 65,000 residents from all demographic sectors and geographic locations of Israel (https://sekernet.co.il/ (accessed on 26 May 2021). A stratified sampling method was employed, aligned with the data published by the Israeli Central Bureau of Statistics, to appropriately include the varied groups of the Israeli Jews population in terms of gender, age, and geographic dispersal. The present study examined negative and positive predictors of psychological coping responses in two different circumstances: the COVID-19 pandemic and the May 2021 hostility. The first data collection was accomplished throughout the third COVID-19 lockdown in Israel, at the beginning of the inoculation operation (14–18 January 2021). Table 1 shows that this sample included 699 participants, 369 men and 330 women, who ranged between 18 and 82 years of age; the income of 50% of them was below the national average, 51% held right-wing political attitudes and 49% of them were secular, and 29% define themselves as traditional. The second data collection was conducted during the 21 May hostility (13–14 May 2021). Table 1 shows that this sample included 647 respondents, 350 men and 297 women, whose ages ranged between 19 and 83 years; the income of 53% of them was below the national average, 52% of them held right-wing political attitudes and 51% of them were secular, and 29% of them define themselves as traditional. Comparing the data between the two samples shows no significant differences between the averages.

### 2.3. Instruments

**Distress**. Two subscales derived from the Brief Symptom Inventory (BSI) [45] were combined into a single distress score. The anxiety subscale consisted of four items referring to felt nervousness, tension, and restlessness. The depression subscale consisted of five items about feelings of worthlessness and hopelessness. Each item was rated on a scale ranging from 1 (not suffering at all) to 5 (suffering very much). Cronbach’s alpha for the distress scale was high (α = 0.90). 

**Well-being**. The well-being scale employed consisted of nine items concerning individuals’ perception of their present lives in various contexts, such as work, family life, health, free time, and others [38]. Responses to these items ranged from 1 (very bad) to 6 (very good). This scale has been validated in previous studies [38], and its reliability in the present study was found to be high (α = 0.85). 

**Individual resilience**. Individual resilience was measured by the 10-item Connor–Davidson scale (CD-RISC 10) [46,47] portraying individual feelings of ability and power in face of difficulties (for example, “I manage to adapt to the changes”). This scale was rated on a 5-point Likert scale ranging from 1 (not true at all) to 5 (generally true). The Cronbach’s alpha reliability of this scale in the present study was high (α = 0.88). 

**Community resilience**. Community resilience was measured by a 10-item scale [48], which was rated by a scale ranging from 1 (does not agree at all) to 5 (very much agrees) (for example, “The relations among the inhabitants of my living place are good”). The current Cronbach’s alpha reliability of this scale was high (α = 0.91). 

**National resilience**. A short version of the National Resilience Scale was employed [31]. This 13-item tool pertained to trust in national leadership, patriotism, and trust in major national institutions (e.g., “I love my country and am proud of it”). In the current study, we added three items regarding the COVID-19 crisis (e.g., “I have full faith in the ability of my country’s health system to care for the population in the current coronavirus crisis”). The 6-point response scale ranged from 1 (very strongly disagree) to 6 (very strongly agree). The Cronbach’s alpha reliability of this scale in the present study was high (α = 0.91).

**Cognitive appraisals**. Each of the four cognitive appraisals was determined by a single item. Health risk: “How much do you feel threatened these days by the health risk?”, economic risk: “How much do you feel threatened these days by the economic risk?”, security risk: “How much do you feel threatened these days by the security risk?”, and political risk: “How much do you feel threatened these days by the political risk?”. The 5-point response scales ranged from 1 (not threatening at all) to 5 (threatening very much). 

**Morale**: Morale was estimated by one item, “How would you define your morale these days?” The response scale ranged from 1 (not good at all) to 5 (very good). 

### 2.4. Data Analysis

Two path analyses in Amos structural equation modeling (IBM, SPSS version 26, https://www.ibm.com/ilen/marketplace/structural-equation-modeling-sem; accessed on 26 May 2021) [49] were utilized to examine our hypotheses. We used maximum likelihood estimates and examined a saturated model, as we did not find any studies that supported an alternative model. It is important to note that in a saturated model, there is no need to examine a model fit as the default and the saturated model are the same [50]. Standardized scores were employed in these path analyses. We repeated the analysis of the routes twice on different samples: the COVID-19 sample and the 21 May hostility sample. The two saturated models (all paths were examined) included five predictors and five predicted psychological coping responses expressed during the two adversities (the COVID-19 pandemic and the 21 May hostility). The predictors were the perceived security, health, economic, and political risks, as well as the level of morale. The predicted variables were the reported levels of distress (composed of both anxiety and depression); individual, community, and national resilience; and well-being. 

## 3. Results

Our first hypothesis claimed that morale, which is a positive cognitive appraisal, will be as good a predictor as the four negative cognitive appraisals (the four different risks) in predicting both positive and negative psychological coping responses during the COVID-19 pandemic and the armed conflict. The first path analysis supported this hypothesis: morale was the best predictor of the levels of distress, well-being, and individual and community resiliencies and was the second-best predictor of national resilience (Table 2) during the COVID-19 pandemic (see also Figure 1, describing the general model of the two path analyses (during COVID-19 and the armed conflict in Gaza)). 

A further examination of this path analysis indicated that, in agreement with our second hypothesis, the perceived health risk (which is supposed to be the most relevant to the pandemic adversity) was not the best predictor of the psychological coping responses expressed during the COVID-19 pandemic. Notice that the perceived health risk was the best predictor of distress and significantly predicted well-being and individual resilience but not community or national resilience. However, it was not a better predictor of well-being than the perceived economic risk, nor was it a better predictor of individual resilience than the security risk. This path analysis showed as well that the perceived political threat was the best predictor of national resilience. The percentages of the variance explained by this path analysis were as follows: distress, 55%; well-being, 48%; individual resilience, 28%; national resilience, 15%; and community resilience, 0.08%.

A second and similar path analysis examined the role of the same predictors in predicting the identical psychological coping responses following a completely different disaster, the May 2021 armed conflict (Table 2 and Figure 1). The percentages of the variance explained by this path analysis were as follows: distress, 59%; well-being, 52%; individual resilience, 33%; national resilience, 17%; and community resilience, 9%. Morale was, again, found to be the best predictor of distress, well-being, and individual and community resilience and the second-best predictor of national resilience. Examination of the role of the perceived security risk (which is supposed to be the most relevant to the security adversity) as a predictor of coping responses, compared to the other three perceived risks, showed that this risk significantly predicted individual and national resilience, as well as distress, but not community resilience or well-being. However, similar to the first path analysis of the COVID-19 pandemic data, this path analysis indicated that the most relevant perceived risk of the investigated adversity was not the best predictor among the four perceived risks in predicting the coping responses. Following the hostility, the perceived security risk did not predict the individual resilience or the distress better than the perceived health risk. Finally, in the 21 May hostility, as well as in the COVID-19 context, the perceived political risk, rather than the morale, was again the best predictor of national resilience. 

## 4. Discussion

The present study examined two major issues that have hardly been studied previously, comparing coping with the stressful experiences of a pandemic versus a security conflict. The first issue referred to the role of positive versus negative cognitive appraisals in predicting psychological coping responses during adversities. The second issue is related to the misconception, according to which the relevant perceived risk of major adversity will constitute the best predictor of the psychological coping responses raised during the disaster. 

The present path analyses confirmed our first hypothesis, showing the major role of positive cognitive appraisals in determining psychological coping. The positive cognitive appraisal of morale constituted a much better predictor of the investigated coping responses than any of the four negative cognitive appraisals employed: perceived security, health, economic, and political risks. The clear advantage of morale as a predictor of most psychological coping responses was retained, due to these path analyses, in four out of the five investigated coping responses along with the COVID-19 pandemic, as well as during the midst of the May 2021 hostility. Positive cognitive appraisals were often regrettably ignored by studies of coping with catastrophes and hardships. We suggest that further research is needed to support the contention that rather than being overlooked, morale and other positive appraisals should indeed be seen as major predictors of these coping processes and, therefore, should be constantly included as variables when studying responses to disasters. Our findings further suggest that efforts to help people reducing negative feelings such as anxiety and depression and to enhance positive feelings, such as individual resilience or well-being, may achieve better results by concentrating on enhancing morale [51] and probably other positive cognitive appraisals, rather than on attempting to reduce their negative cognitive appraisals, including various perceived threats. 

To date, most of the available research on predicting psychological responses to adversities concentrated on the negative responses to threatening conditions. A study of the SARS pandemic has thus reported that anxiety, avoidance, and disgust were the aftermath of the threats of this plague [20]. A more recent study concluded that higher levels of anxiety and depression, as well as lower levels of well-being and individual resilience, reflected the perceived risk of the COVID-19 pandemic [38]. Our contention that psychological coping responses to adversity will also be affected by positive cognitive appraisals was supported by a different perspective, which argued that the levels of depression and distress of victims of earthquake and tsunami decreased as a result of proactive appraisals [23]. This perspective was further supported by a comprehensive analysis of psychological reactions to wars [26], according to which distress casualties in military units did not reflect the perceived war threats. Units with a higher level of morale suffered a substantially lower number of distress injuries compared to troops with a lower level of morale.

A large number of studies of adversities and disasters tend to take for granted the quite reasonable conclusion that the increased negative psychological responses, which follow such disasters, as well as the ensuing lower levels of hope and resilience, reflect the perceived impact of the investigated calamity. Yang and Ma [6] have thus concluded that the current decrease in the overall emotional well-being in China represents a negative response to the perceived threat of the COVID-19 pandemic. Levkovich and Shinan-Altman [10] have similarly reported that the perceived COVID-19 pandemic threat explains the negative emotional reactions expressed by the general Israeli public during this plague. A study of several wars has concluded, by the same token, that the fear of armed conflicts is responsible for higher levels of anxiety, depression, and PTSD [11]. However, these studies did not compare the perceived impact of their investigated hardships with the impacts of other perceived threats, such as economic difficulties, discrimination, or political concerns, on the psychological coping responses, which were expressed in association with each of these calamities. 

The present study questioned the contention that coping responses can mostly be explained by the direct impact of the perceived threats of an investigated disaster. This issue was investigated by comparing the relative contributions of the most relevant perceived risk of the pandemic (heath risk) or the security conflict (security risk) in predicting coping responses with the predictions of the other three less relevant perceived risks. Our results refute the claim that these coping responses reflect mainly the most relevant perceived threat of each of these two calamities. The first path analysis showed, in line with our second hypothesis, that the perceived health risk was the best predictor of distress but was not the best predictor of the rest of the psychological coping responses, expressed during the COVID-19 pandemic. Similarly, the second path analysis indicated that the perceived security risk best predicted the level of distress expressed following the 21 May conflict but was not the best predictor of any of the other coping responses. 

### Limitations

A major limitation of this study is the fact that it is based on subjective self-reports of the general public. Studies of this kind are dependent on the respondents’ awareness of their perceived levels of risk, which were raised by the four potential threats, the extent to which they feel anxious or resilient during these adversities. Therefore, this subjective data collection method always raises the question of whether the reports accurately represent all these psychological responses. Another limitation is the use of an internet panel company, which has a similar problem as sampling in the general population. Only those willing to take part are included. They may have systematic differences from those not willing to take part that might be relevant to the current study. The third limitation is that the study was conducted only among the Israeli Jews (the majority) population, due to budget constraints (distributed in Hebrew only). Thus, additional research is required to test the generalizability of the findings to other populations.

## 5. Conclusions

Two main findings were found in the current study. One relates to the importance of morale as a predictor of psychological coping with distress in varied types of adversities. The second applies to the need to consider varied indirect perceived risks that impact populations during crises, rather than the tendency, which characterizes many previous studies, to focus only on the most apparent or “expected” type of risk. 

Many studies focus on disasters to identify either the sources or the predictors of different psychological coping responses expressed during such periods. Our findings strongly suggest that the variances of coping responses explained by these studies may be substantially increased by including positive cognitive appraisals, such as morale, among their predictors. This conclusion is based on the fact that the present study employed morale as such a predictor and found that in most cases it predicted coping responses better than most of the negative perceived cognitive appraisals. A higher level of morale was correlated with an increased sense of well-being, as well as higher individual and community resilience. At the same time, it was also correlated with reduced levels of distress. It seems reasonable to expect that other positive appraisals may contribute further to predicting these responses.

Our second conclusion pertains to the issue of a single major predictor of coping responses in the case of a catastrophe versus a model with multiple predictors. A large number of studies have taken for granted the assumption that negative and positive coping responses are caused directly by the relevant perceived threat of the investigated catastrophe. However, our findings present a more complicated phenomenon. First, all the investigated coping responses were predicted much better by the level of morale than by the perceived risks of the investigated adversities. Second, the perceived health risk and the perceived security risk were the best predictors of a single coping response, and the other perceived risks predicted better the rest of the coping responses. We suggest that overgeneralization, in which all the coping responses that are revealed during specific jeopardy are accounted for by the perceived risk of the catastrophe, should be avoided in future studies of adversities. It is strongly recommended that future studies of this issue empirically test this assumption by examining the contribution of additional potential predictors of coping responses, especially more indirect perceived risks.

## Figures and Tables

**Figure 1 ijerph-18-08759-f001:**
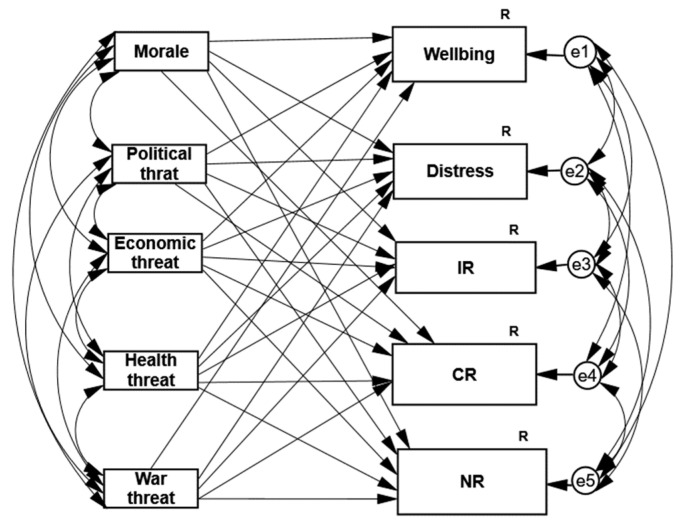
General model of the two path analyses—four types of threats and morale predicting distress; well-being; and individual, community, and national resilience during COVID-19 and Gaza armed conflict. R = square multiple correlations (explained variability in percentage). The letters and numbers represent residual error with ID number.

**Table 1 ijerph-18-08759-t001:** Demographic characteristics of the two investigated samples: COVID-19 (699) and Gaza 21 May conflict (647).

Variable		COVID-19			Armed Conflict		
		*n*	%	M (SD)	*n*	%	M (SD)
Age groups	18–30	127	18	45.67 (15.24)	95	15	48.52 (26.79)
	31–40	168	24		145	22	
	41–50	138	20		136	21	
	51–60	123	18		116	18	
	≥61	143	21		152	24	
Gender	Women	330	47		297	54	
	Men	369	53		350	46	
Income level	Below	346	50	2.56 (1.22)	339	53	2.48 (1.26)
	Average	177	25		154	24	
	Above	176	25		54	24	
Political attitudes	Left	87	13	3.47 (0.84)	77	11	3.48 (1.70)
	Center	249	36		236	37	
	Right	363	51		334	52	
Religiosity	Secular	345	49	1.79 (0.95)	328	51	1.79 (0.96)
	Traditional	203	29		177	27	
	Religious	94	13		91	14	
	Orthodox	57	8		51	8	

**Table 2 ijerph-18-08759-t002:** Standardized estimates of path analyses of four types of threats and morale predicting distress; well-being; and individual, community, and national resilience during COVID-19 (*n* = 699) and Gaza 21 May conflict (*n* = 647).

Predictor	Predicted	COVID Estimate	Gaza 21 May Conflict Estimate
Morale	WB	0.53 ***	0.53 ***
	Distress	−0.58 ***	−0.56 ***
	IR	0.44 ***	0.42 ***
	CR	0.23 ***	0.18 ***
	NR	0.20 ***	0.24 ***
Political threat	WB	−0.05	0.01
	Distress	0.03	0.01
	IR	0.03	0.09
	CR	−0.02	0.05
	NR	−0.28 ***	−0.27 ***
Economic threat	WB	−0.18 ***	−0.14 ***
	Distress	0.11 ***	0.11 ***
	IR	−0.05	0.01
	CR	−0.04	−0.11 **
	NR	−0.13 **	−0.08 *
Health threat	WB	−0.15 ***	−0.25 ***
	Distress	0.12 ***	0.09 **
	IR	0.07	0.23 ***
	CR	0.03	−0.03
	NR	0.08	0.04
Security threat	WB	0.04	0.07
	Distress	0.11	0.20 ***
	IR	0.12 **	0.09 *
	CR	0.03	−0.07
	NR	0.10 *	0.11 *
Explained variance R^2^	WB	0.47	0.52
	Distress	0.63	0.59
	IR	0.28	0.33
	CR	0.08	0.09
	NR	0.06	0.17

WB = well-being, IR = individual resilience, CR = community resilience, NR = national resilience. * *p* < 0.05, ** *p* <0.01, *** *p* < 0.001.

## Data Availability

All data accumulated in the study are available to the authors. Data are not published openly due to privacy issues, but analyzed data are available from the authors upon request.

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
