# Peer review of "Morale and Perceived Threats as Predictors of Psychological Coping with Distress in Pandemic and Armed Conflict Times"

_ijerph, 2021, doi:10.3390/ijerph18168759_

Round 1

Reviewer 1 Report

This study raises important questions. Most research focuses on the direct link between the event and the response, bringing in other factors as mediators or moderators. This study changes that focus, providing evidence that the alreday existing state of the population has a greater impact than the event in terms of predictive factors. This is a key finding, and the article should emphasise this aspect more clearly.

There are some minor issues

  • clarify findings in the Abstract
  • 111 - Israel also attacked Gaza. This needs to be noted as it will have an impact on the study, eg morale may be increased by perceived effective defences, alternatively it may be decreased by perceived inappropriate attacks on Gaza.
  • 188 - it is looking like there are further waves of Covid so this may be changed to take this into account.
  • 199 - Internet panel company, still has the same problems as sampling in the general population. Only those willing to take part are included. They may have systematic differences from those not willing to take part that might be relevant to the current study. Note this in limitations.

Overall I think this is publishable. The presentation could at times be improved to eg emphasise the key findings and conclusions. There should be more emphasis on how this approach differs from how research similar to this is usually conducted.

Author Response

We thank Reviewer 1 for his/her comments. We've revised the manuscript accordingly. Please see below the specific answer to each comment:

  • 111 - Israel also attacked Gaza. This needs to be noted as it will have an impact on the study, eg morale may be increased by perceived effective defences, alternatively it may be decreased by perceived inappropriate attacks on Gaza.

Response: We've added this to the manuscript; see lines 115-116

  • 188 - it is looking like there are further waves of Covid so this may be changed to take this into account.

Response: This was added to the manuscript; See lines193-196

  • 199 - Internet panel company, still has the same problems as sampling in the general population. Only those willing to take part are included. They may have systematic differences from those not willing to take part that might be relevant to the current study. Note this in limitations.

Response: this limitation was added to the limitations section. See lines 396-399

The presentation could at times be improved to eg emphasise the key findings and conclusions. There should be more emphasis on how this approach differs from how research similar to this is usually conducted.

Response: We emphasized the key findings and how they differ from previous studies. See lines 404-408

Reviewer 2 Report

The authors describe an interesting study of the predictors of psychological coping, and draw an important conclusion that morale improved the predictions of the varied coping responses in both the pandemic and conflict setting. As such, it provides a useful contribution to the field. The manuscript also provides a very nice review of the relevant literature.

The following suggestions could improve the manuscript.

Line 36: …..and a range of other NEGATIVE mental health outcomes  (or, detrimental?)

Line 170-171:…morale, which constitutes a positive cognitive appraisal, will be AS a good A predictor as the four investigated negative cognitive appraisals….

In table 2, please define the meaning of the asterisks. Also, Standardized estimatesd ….

Figure 1: In the legend, define R and the numbering and lettering on the right side

Please be consistent and neutral with language: Line 114 refers to ‘Arab riots’ but line 196 refers to ‘riots between Arabs and Jews’. I also found myself wondering why only the ‘Israeli Jews population’ was studied (line 203).  This warrants some explanations for the selected study population. Also, how does this affect the generalizability of the findings to other populations?

The authors suggest that “efforts to help people reducing negative feelings such as anxiety and depression and to enhance positive feelings, such as individual resilience or well-being, may achieve better results by concentrating on enhancing morale”, but there is no citation of evidence to indicate that there are effective interventions to ‘enhance morale’  (Line 338)

Author Response

We thank Reviewer 2 for his wise comments. We revised the manuscript accordingly. Please see below the specific answers to each comment:

Line 36: …..and a range of other NEGATIVE mental health outcomes  (or, detrimental?)

Response: this was revised as suggested. See lines 36-37

Line 170-171:…morale, which constitutes a positive cognitive appraisal, will be AS a good A predictor as the four investigated negative cognitive appraisals….

Response: this was revised as suggested. See lines 173

In table 2, please define the meaning of the asterisks. Also, Standardized estimatesd ….

Response: we added the meaning of the asterisks. See table 2, line 291; modified 'estimates'. See line 287

Figure 1: In the legend, define R and the numbering and lettering on the right side

Response: Figure 1 was re-done, to include description of what R, numbering and letters signify. See lines 296-297

Please be consistent and neutral with language: Line 114 refers to ‘Arab riots’ but line 196 refers to ‘riots between Arabs and Jews’.

Response: this was revised accordingly to maintain consistency. See lines 114-115

I also found myself wondering why only the ‘Israeli Jews population’ was studied (line 203).  This warrants some explanations for the selected study population. Also, how does this affect the generalizability of the findings to other populations? 

Response: the study was conducted only among Israeli Jews (the majority) population due to budget constraints (distributed in Hebrew only). This was added to the limitation section, including the reference to the generalizability of the findings to other populations. See lines 402-405

The authors suggest that “efforts to help people reducing negative feelings such as anxiety and depression and to enhance positive feelings, such as individual resilience or well-being, may achieve better results by concentrating on enhancing morale”, but there is no citation of evidence to indicate that there are effective interventions to ‘enhance morale’  (Line 338)

Response: A relevant reference was added concerning the ability of using effective interventions to enhance morale. See lines 350 and 490-491